# Conditional Matrix Flows for Gaussian Graphical Models

**Marcello Massimo Negri**
University of Basel
marcellomassimo.negri@unibas.ch

**Fabricio Arend Torres**
University of Basel
fabricio.arendtorres@unibas.ch

**Volker Roth**
University of Basel
volker.roth@unibas.ch

## Abstract

Studying conditional independence among many variables with few observations is a challenging task. Gaussian Graphical Models (GGMs) tackle this problem by encouraging sparsity in the precision matrix through $l_q$ regularization with $q \leq 1$. However, most GMMs rely on the $l_1$ norm because the objective is highly non-convex for sub-$l_1$ pseudo-norms. In the frequentist formulation, the $l_1$ norm relaxation provides the solution path as a function of the shrinkage parameter $\lambda$. In the Bayesian formulation, sparsity is instead encouraged through a Laplace prior, but posterior inference for different $\lambda$ requires repeated runs of expensive Gibbs samplers. Here we propose a general framework for variational inference with matrix-variate Normalizing Flow in GGMs, which unifies the benefits of frequentist and Bayesian frameworks. As a key improvement on previous work, we train with one flow a continuum of sparse regression models jointly for all regularization parameters $\lambda$ and all $l_q$ norms, including non-convex sub-$l_1$ pseudo-norms. Within one model we thus have access to (i) the evolution of the posterior for any $\lambda$ and any $l_q$ (pseudo-) norm, (ii) the marginal log-likelihood for model selection, and (iii) the frequentist solution paths through simulated annealing in the MAP limit.

## 1 Introduction

Estimating complex relationships between random variables is a central problem in science. When only few observations are available, the problem becomes even more challenging. Examples include functional connectivity in fMRI data [Smith et al., 2013], networks of interactions from microarray data [Castelo and Roverato, 2006], or correlation patterns in longitudinal studies [Diggle, 2002]. In such cases, it is particularly challenging to infer the conditional independence among random variables. Graphical models are commonly used to represent such an independence structure in the form of network graphs. In this work, we focus specifically on Gaussian Graphical Models (GGMs). GGMs assume observations $\boldsymbol{X} \in \mathbb{R}^{n \times d}$ to be generated from a multivariate Gaussian distribution $\mathcal{X} \sim \mathcal{N}(\boldsymbol{\mu}, \boldsymbol{\Sigma})$ with $\boldsymbol{X}_{i,:}$ being realizations of $\mathcal{X}$ for $i \in \{1, \ldots, n\}$. Here $\boldsymbol{\mu} \in \mathbb{R}^d$ denotes the mean and $\boldsymbol{\Sigma} \in \mathbb{R}^{d \times d}$ the covariance matrix, which is symmetric positive definite $\boldsymbol{\Sigma} \succ 0$. GGMs provide a simple interpretation of conditional independence through the precision matrix $\boldsymbol{\Omega} = \boldsymbol{\Sigma}^{-1}$, whenever $\boldsymbol{\Sigma}$ is non-singular. Specifically, $\boldsymbol{\Omega}_{i,j} = 0$ implies that the pair of variables $(i, j)$ is conditionally independent given all remaining variables. That is to say, there is no edge between nodes $i$ and $j$ in the underlying undirected graph.

37th Conference on Neural Information Processing Systems (NeurIPS 2023).

**Penalized likelihood formulation** Given the centered observations $\boldsymbol{X}$ and the associated sample covariance matrix $\boldsymbol{S} = \boldsymbol{X}^T \boldsymbol{X}$, we are interested in reconstructing the precision matrix $\boldsymbol{\Omega}$, hence the underlying graph structure. This is particularly challenging when $d > n$ because the sample covariance matrix becomes singular and the precision matrix can no longer be obtained by simply inverting the MLE of the covariance matrix. One way to overcome this problem is to consider likelihood-penalized models that encourage sparsity in the precision matrix. In other words, we trade off the likelihood with the number of zeros in $\boldsymbol{\Omega}$:

$$\hat{\boldsymbol{\Omega}} = \arg\max_{\boldsymbol{\Omega} \succ 0}\{\log \det \boldsymbol{\Omega} - \mathrm{Tr}(\tfrac{1}{2}\boldsymbol{S}\boldsymbol{\Omega}) - \lambda\|\boldsymbol{\Omega}\|_0\}\,, \tag{1}$$

where $\log \det \boldsymbol{\Omega} - \mathrm{Tr}(\tfrac{1}{2}\boldsymbol{S}\boldsymbol{\Omega})$ is the log-likelihood term and $\|\boldsymbol{\Omega}\|_0 = \sum_{i<j} 1[w_{ij} \neq 0]$ is the $l_0$ norm, which counts the number of non-zero off-diagonal elements of $\boldsymbol{\Omega}$. The trade-off between the two terms is controlled by the parameter $\lambda \geq 0$. Note that the optimization must be performed over the space of symmetric positive definite matrices $\boldsymbol{\Omega} \succ 0$. In practice, the objective in Eq. (1) is highly non-convex and cannot be optimized easily. Even in the simpler linear regression setting, $l_0$ regularization can be computed exactly only for few features [Hastie et al., 2015]. Similar problems arise for all non-convex sub-$l_1$ pseudo-norms.

**Lasso relaxation: Frequentist and Bayesian approaches** In order to simplify Eq. (1), most approaches replace the highly non-convex $l_0$ norm with the closest convex norm, the $l_1$ norm: $\|\boldsymbol{\Omega}\|_1 = \sum_{i<j} |w_{ij}|$. Meinshausen and Bühlmann [2006] first proposed using $l_1$ regularization in the Graphical Lasso model, which sparked the development of several likelihood penalized algorithms [Friedman et al., 2007, Yuan and Lin, 2007, Banerjee et al., 2007]. These frequentist approaches make it possible to elegantly compute the solution path as a function of the shrinkage parameter $\lambda$ [Mazumder and Hastie, 2012]. The $l_1$-relaxed problem can also be easily extended to a Bayesian formulation. Wang [2012] assumed a Laplace prior on the off-diagonal elements of the precision matrix and showed that the frequentist solution can be recovered as the maximum a posteriori estimate (MAP). In the Bayesian framework, it is possible to explore the full posterior distribution, to formulate the posterior predictive, and to use the marginal likelihood for model selection. But in high-dimensional settings, MCMC samplers suffer from poor mixing behaviour, which becomes increasingly difficult to diagnose in higher dimensions. Furthermore, different hyperparameters values, e.g., $\lambda$, require independent expensive Markov chains. In the literature, alternative priors have also been proposed [Li et al., 2019]. However, the dependence of Gibbs samplers on tractable posterior conditionals typically restricts the choice of priors significantly.

**Variational Inference** Variational inference approaches [Blei et al., 2017] approximate intractable densities with tractable parameterized distributions. Compared to MCMC samplers, variational inference turns a sampling problem into an optimization one, which generally requires less computational time. Let $\boldsymbol{x}$ be the observed variables and $\boldsymbol{z}$ the unobserved ones. Given a family $\mathcal{Q} = \{q_\theta(\boldsymbol{z})|\theta \in \Theta\}$ of parameterized distributions, variational inference involves finding the distribution $q_{\theta*}(\boldsymbol{z})$ that best approximates the posterior $p(\boldsymbol{z}|\boldsymbol{x})$. The distance between the two distributions is usually measured as the Kullback-Leibler divergence, which defines the following optimization problem:

$$q_{\theta*}(\boldsymbol{z}) = \arg\min_{\theta \in \Theta} \mathrm{KL}\big(q_\theta(\boldsymbol{z})||p(\boldsymbol{z}|\boldsymbol{x})\big) = \arg\min_{\theta \in \Theta} \mathbb{E}_{\boldsymbol{z} \sim q_\theta}\left[\log \frac{q_\theta(\boldsymbol{z})}{p(\boldsymbol{z}|\boldsymbol{x})}\right]. \tag{2}$$

The goodness of $q_{\theta*}(\boldsymbol{z})$ as an approximation of the posterior $p(\boldsymbol{z}|\boldsymbol{x})$ creates a trade-off between the expressive power of the variational family $\mathcal{Q}$ and the tractability of the optimization in Eq. (2). Common approaches rely on the so-called mean field approximation, which assumes mutual independence among variables: $q_\theta(\boldsymbol{z}) = \prod_i q_{\theta_i}(z_i)$. But this is not viable in GGMs as we intend to model precisely the dependence structure. Several approaches have been proposed for Bayesian Lasso [Alves et al., 2021] and for its group-sparse variant [Babacan et al., 2014] Yet, to the best of our knowledge, variational approaches have not been studied in the context of Bayesian GGMs.

**Contribution** We present a unified approach to sparse regression for the whole family of all $l_q$ (pseudo-) norms with $0 < q < \infty$, which allows for a fully Bayesian and frequentist-type interpretation. Specifically, we propose a very general framework for variational inference in Bayesian GGMs through a Normalizing Flow defined over the space of symmetric positive definite matrices. By conditioning the flow on $\lambda$ and on $q$, we simultaneously train a continuum of sparse regression

models for all choices of shrinkage parameters $0 < \lambda < \lambda_{\max}$ and all $l_q$ (pseudo-) norms with $0 < q < q_{\max}$, including the highly non-convex sub $l_1$ pseudo-norms. We use as prior the generalized Normal distribution, which for $q = 1$ recovers the Laplace prior and the Lasso regularization in the MAP limit. On the one hand, our approach inherits the advantages of the Bayesian framework while altogether avoiding problems arising from Gibbs sampling strategies. On the other hand, we can still recover the frequentist solution path in the MAP limit by training through simulated annealing.

In summary, our main contributions are the following:

1. We propose a general framework for variational inference in GGMs through a normalizing flow defined directly over the space of symmetric positive definite matrices. We condition such a flow on the shrinkage parameter $\lambda$ and on $q > 0$ to model $l_q$ (pseudo-) norms.

2. We combine the advantages of the frequentist and Bayesian frameworks: in a single model we have access to posterior inference and to the marginal likelihood as a function of $\lambda$ and $q$. We can further recover the frequentist solution path as the MAP by annealing the system.

3. To the best of our knowledge, we are the first to propose a unified framework for sub-$l_1$ pseudo-norms that does not require surrogate penalties and that enables consistent exploration of the full solution paths in terms of $\lambda$ and $q$.

## 2   Related Work

In this section, we provide an overview of the Bayesian Graphical Lasso and illustrate the limitations of current inference methods. We then outline the limitations of Lasso regularization and briefly review existing approaches for sparsity with sub-$l_1$ pseudo-norms. Lastly, as an alternative to MCMC approaches, we briefly review variational inference with normalizing flows.

**Bayesian Graphical Lasso**    Similarly to the Bayesian Lasso [Park and Casella, 2008], Wang [2012] provided a Bayesian interpretation of the Graphical Lasso (BGL) for posterior inference. Specifically, they showed that the $l_1$-relaxed optimization of Eq. (1) is equivalent to the maximum a posteriori estimate (MAP) of the model defined by $p(\boldsymbol{\Omega}|\boldsymbol{S}, \lambda) \propto p(\boldsymbol{S}|\boldsymbol{\Omega})\, p(\boldsymbol{\Omega}|\lambda)$, with

$$p(\boldsymbol{S}|\boldsymbol{\Omega}) \propto \mathcal{W}_d(n, \boldsymbol{\Omega}^{-1})$$
$$p(\boldsymbol{\Omega}|\lambda) \propto \mathcal{W}_d(d + 1, \lambda \mathbf{1}_d) \prod_{i<j} \mathrm{DE}(\omega_{ij}|\lambda) I[\boldsymbol{\Omega} \succ 0] \,, \tag{3}$$

where the indicator function $I[\boldsymbol{\Omega} \succ 0]$ imposes positive definiteness on $\boldsymbol{\Omega}$. The likelihood term reflects that the sample covariance matrix $\boldsymbol{S} = \boldsymbol{X}^T \boldsymbol{X}$ is distributed according to the Wishart distribution $\mathcal{W}_d(n, \boldsymbol{\Omega}^{-1}) \propto \det(\boldsymbol{\Omega})^{n/2} \det(\boldsymbol{S})^{(n-d-1)/2} \exp \mathrm{Tr}(-\frac{1}{2}\boldsymbol{\Omega}\boldsymbol{S})$. The prior is instead composed of two terms. The Wishart term $\mathcal{W}_d(d + 1, \lambda \mathbf{1}_d)$ imposes symmetry and positive definiteness on $\boldsymbol{\Omega}$. In addition, the double exponential (or Laplace) prior $\mathrm{DE}(\omega_{ij}|\lambda) = \frac{\lambda}{2} \exp\{-\lambda|\omega_{ij}|\}$ encourages sparsity on the off-diagonal elements of $\boldsymbol{\Omega}$.

Inference in the BGL, and more generally for Bayesian Lassos, is performed through Gibbs samplers. However, these MCMC strategies are computationally expensive and can suffer from high rejection rates, especially in high dimensions [Mohammadi and Wit, 2015]. Furthermore, to recover the frequentist solution path in the MAP limit, the Markov Chain must be restarted for each $\lambda$ value. Its derivation also requires to expand the Laplace prior as an infinite mixture of Gaussians [Andrews and Mallows, 1974, West, 1987] and to define an ad hoc mixing density. Lastly, the obtained Gibbs sampler does not generalize to other priors.

**Sparsity with sub-$l_1$ pseudo-norms**    The $l_0$ norm is particularly suited to enforce sparsity because it directly penalizes the non-zero entries. In other words, it is equivalent to best subset selection of the variables. Convex relaxation with the $l_1$ norm makes the problem tractable, but comes at the cost of encouraging shrinkage also on the retained variables [Hastie et al., 2015]. This motivated interest in sub-$l_1$ pseudo-norms, which in the limit reduce to the original $l_0$ norm formulation. Due to their non-convexity and combinatorial complexity, multiple surrogate objectives have been proposed. Notably, with the proposed approach we do not have to resort to alternative objectives or penalties, as we can directly and exactly enforce sub-$l_1$ pseudo-norm regularization with a suitable prior. In the context of linear regression, several algorithms have been proposed. The popular SCAD [Fan and Li,

2001] guarantees unbiasedness, sparsity, and continuity while reducing the overshrinking behaviour. Zhang [2010] proposed the nearly unbiased MC+ method, which consists of a concave penalty and a selection algorithm that bridges the gap between $l_1$ and $l_0$. In the literature, other penalties that approximate the $l_0$ penalty have been proposed, such as the SICA penalty [Lv and Fan, 2009], the log-penalty [Mazumder et al., 2011], the seamless-$l_0$ penalty [Dicker et al., 2013] and the atan penalty [Wang and Zhu, 2016]. In the Bayesian framework, Ishwaran and Rao [2005] used a rescaled spike and slab prior to encourage variable selection and drew connections to Ridge regularization.

**Normalizing Flows for variational inference**  Normalizing Flows (NFs) are flexible models that can perform accurate density estimation For this reason, NFs represent a very attractive solution to perform efficient and accurate variational inference [Rezende and Mohamed, 2015, van den Berg et al., 2019]. Suppose $\mathcal{X}$ is a continuous $d$-dimensional random variable with unknown distribution $p_{\boldsymbol{x}}$, and let $\mathcal{Z}$ be any continuous $d$-dimensional random variable with some known base distribution $\boldsymbol{z} \sim p_{\mathcal{Z}}$. The key idea of NFs is to construct a diffeomorphism $\mathcal{T} : \operatorname{supp}(\mathcal{X}) \mapsto \operatorname{supp}(\mathcal{Z})$, i.e., a differentiable bijection, in order to rewrite $p_{\mathcal{X}}$ through the change of variable formula as

$$p_{\mathcal{X}}(\boldsymbol{x}) = p_{\mathcal{Z}}\big(\mathcal{T}(\boldsymbol{x})\big) \left| \det \mathcal{J}_{\mathcal{T}}(\boldsymbol{x}) \right| \ , \tag{4}$$

where $\mathcal{J}_{\mathcal{T}}$ is the Jacobian of the transformation $\mathcal{T}$. Assuming that we have successfully learned the transformation $\mathcal{T}$, we can then evaluate the target density through Eq. (4). We can further sample from the target distribution by simply transforming samples of the base distribution through $\mathcal{T}$, namely $\boldsymbol{x} = \mathcal{T}(\boldsymbol{z})$ with $\boldsymbol{z}$ being realizations of $p_{\mathcal{Z}}$. In practice, the crucial part of designing NFs is to construct arbitrarily complicated bijections. To do so, NFs exploit that compositions $\mathcal{T} = \mathcal{T}_1 \circ \cdots \circ \mathcal{T}_m$ of bijections $\{\mathcal{T}_i\}_{i=1}^{m}$ remain bijections. The determinant of the resulting transformation decomposes into the determinants of each $\mathcal{T}_i$ as

$$\det \mathcal{J}_{\mathcal{T}}(\boldsymbol{x}) = \prod_{i=1}^{m} \det \mathcal{J}_{\mathcal{T}_i}(\boldsymbol{u}_{i-1}) \quad \text{with} \quad \boldsymbol{u}_{i-1} = \mathcal{T}_1(\boldsymbol{x}) \circ \cdots \circ \mathcal{T}_{i-1}(\boldsymbol{u}_{i-2}) \ . \tag{5}$$

Therefore, by composing computationally tractable non-linear bijections, it is possible to define arbitrarily expressive bijections and hence to transform the base distribution $p_{\mathcal{Z}}$ into arbitrarily complicated distributions $p_{\mathcal{X}}$. Among others, Huang et al. [2018] proved that autoregressive flows are universal density approximators of continuous random variables. For a comprehensive review of NFs and their different architectures, see Papamakarios et al. [2021] and Kobyzev et al. [2020]. As first proposed by Atanov et al. [2020], it is also possible to condition the transformation $\mathcal{T}$ on some parameter $c \in \mathbb{R}^n$. The resulting Conditional NF [Kobyzev et al., 2020] models with one flow the family of conditional distributions $p_{\mathcal{X}}(\boldsymbol{x}|c)$ for continuous values of $c$.

## 3  Proposed approach

In this section, we describe how to infer the posterior of $\boldsymbol{\Omega}$ in GGMs with conditional NFs and how to do so as a function of the shrinkage parameter $\lambda$ and of the $l_q$ (pseudo-) norm regularization, all within a single model. We illustrate how to define the conditional flow directly over the space of positive definite matrices and argue why training and posterior inference is particularly efficient. Furthermore, we show that NFs provide direct access to the marginal log-likelihood for model selection and show how to recover the frequentist solution path by training through simulated annealing. We term the resulting model Conditional Matrix Flow (CMF).

### 3.1  Generalized Normal distribution for $l_q$ (pseudo-) norms

Unlike previous Bayesian approaches to GGMs, we can flexibly specify any prior (and likelihood) in Eq. (3) without worrying about the existence of a suitable Gibbs sampler. We exploit such flexibility to extend the model beyond the standard $l_1$ relaxation. Wang [2012] showed that in the MAP limit $l_1$ regularization is recovered with a Laplace prior on the off-diagonals of $\boldsymbol{\Omega}$. We generalize this idea to any $l_q$ (pseudo-) norm with the generalized Normal distribution as prior, which reduces to Laplace and Normal distributions for $q = 1$ and $q = 2$, respectively. Its probability density is defined as:

$$f(x|\alpha, \beta) = \frac{\beta}{2\alpha\Gamma(1/\beta)} \exp\left\{ -\frac{|x|^\beta}{\alpha^\beta} \right\}, \tag{6}$$

where $\Gamma(\cdot)$ is the Gamma function, $\alpha, \beta > 0$ are the scale and shape parameters, respectively. Here, we assumed the distribution to be centered around zero, i.e. $\mu = 0$. In order to link the prior to the $l_q$ norm, we rename its parameters as $\lambda = \alpha^{-\beta} > 0$ and $q = \beta$. Our full model can now be conditioned through the prior on both $\lambda$ and $q$ and is defined as $p(\mathbf{\Omega}|\mathbf{S}, \lambda, q) \propto p(\mathbf{S}|\mathbf{\Omega}) \, p(\mathbf{\Omega}|\lambda, q)$ with

$$p(\mathbf{S}|\mathbf{\Omega}) \propto \mathcal{W}_d(n, \mathbf{\Omega}^{-1})$$

$$p(\mathbf{\Omega}|\lambda, q) \propto \mathcal{W}_d(d+1, \lambda\mathbf{1}_d) \prod_{i<j} \frac{q\lambda^{1/q}}{2\Gamma(1/q)} \exp\{-\lambda|\omega_{ij}|^q\} I[\mathbf{\Omega} \succ 0] \,. \tag{7}$$

If we now consider the log-probability and drop the normalization constant, which does not affect the KL optimization, we recover the $l_q$ (pseudo-) norm. For $q = 1$ the prior on the off-diagonals reduces to the Laplace distribution and we recover the Bayesian Graphical Lasso in Eq. (3). For $q = 2$ we recover instead the normal distribution, i.e., Ridge regularization. Overall, when employing the distribution in Eq. (7), we can effectively model solutions corresponding to $l_q$ (pseudo-) norm regularization for $q > 0$, including the non-convex sub-$l_1$ pseudo-norms.

## 3.2 Variational inference with conditional flows for GGMs

As we want to model the posterior $p(\mathbf{\Omega}|\mathbf{S})$ over the precision matrix $\mathbf{\Omega}$, we design flows directly over the space of symmetric positive definite matrices. Specifically, we train a NF conditioned on $\lambda \in [\lambda_1, \lambda_2]$ and on $q \in [q_1, q_2]$ with $\lambda_1, \lambda_2, q_1, q_2 > 0$ in order to study the evolution of the posterior as a function of $\lambda$ and $q$. From Eq. (4) we see that the resulting flow $\mathcal{T}_{\boldsymbol{\theta}(\lambda, q)}$ implicitly defines the probability $q_{\boldsymbol{\theta}(\lambda, q)}(\mathbf{\Omega}) = p_{base}(\mathcal{T}_{\boldsymbol{\theta}(\lambda, q)}(\mathbf{\Omega}))|\det \mathcal{J}_{\mathcal{T}_{\boldsymbol{\theta}(\lambda, q)}}(\mathbf{\Omega})|$, which we want to train to approximate the posterior $p(\mathbf{\Omega}|\mathbf{S}, \lambda, q) \propto p(\mathbf{S}|\mathbf{\Omega}) \, p(\mathbf{\Omega}|\lambda, q)$. We minimize the KL divergence in Eq. (2) and approximate the expectation with Monte Carlo samples:

$$\mathcal{L}(\boldsymbol{\theta}; \lambda, q) = \mathrm{KL}\big(q_{\boldsymbol{\theta}(\lambda, q)}(\mathbf{\Omega})||p(\mathbf{\Omega}|\mathbf{S}, \lambda, q)\big)$$

$$\approx \frac{1}{M} \sum_{i=1}^{M} \log \frac{q_{\boldsymbol{\theta}(\lambda, q)}(\mathbf{\Omega}_i)}{p(\mathbf{S}|\mathbf{\Omega}_i) \, p(\mathbf{\Omega}_i|\lambda, q)} + N_C(\lambda, q). \tag{8}$$

Note that this is particularly efficient in NFs because it only requires sampling from the base distribution $p_{base}(\mathcal{T}_{\boldsymbol{\theta}(\lambda, q)}(\mathbf{\Omega}))$, which is computationally cheap. We further need to evaluate the unnormalized posterior $p(\mathbf{\Omega}|\mathbf{S}, \lambda, q) \propto p(\mathbf{S}|\mathbf{\Omega}) \, p(\mathbf{\Omega}|\lambda, q)$, i.e. the product of the likelihood and the prior, which is also trivial. Relevantly, in Eq. (8) we do not need to compute the normalization constant $N_C(\lambda, q) = \log p(\mathbf{S}|\lambda, q)$ because it does not influence the optimization.

Unlike standard Bayesian approaches, the proposed conditional flow provides access directly to the marginal log-likelihood $N_C(\lambda, q) = \log p(\mathbf{S}|\lambda, q)$ as a function of $\lambda$ and $q$ without further calculations. This is particularly interesting because the marginal log-likelihood is extremely expensive to compute with classical approaches. If we assume that the flow is expressive enough and that the optimization reached the global minimum, then $\mathrm{KL}\big(q_{\boldsymbol{\theta}(\lambda, q)}(\mathbf{\Omega})||p(\mathbf{\Omega}|\mathbf{S}, \lambda, q)\big) \approx 0$. In this case, the loss function at the end of training gives us access directly to the negative marginal log-likelihood for continuous values of $\lambda$ (and $q$). We can then perform model selection by simply choosing $\lambda^* = \max_\lambda \log p(\mathbf{S}|\lambda, q)$.

## 3.3 Conditional Matrix Flow

We now describe the proposed Conditional Matrix Flow (CMF) and how we define it over the space of symmetric positive definite matrices by construction. We exploit the well-known Cholesky decomposition [Horn and Johnson, 1985], which states that any symmetric positive definite matrix $\mathbf{M} \in \mathbb{R}^{d \times d}$ can be decomposed as $\mathbf{M} = \mathbf{L}\mathbf{L}^T$, where $\mathbf{L} \in \mathbb{R}^{d \times d}$ is a lower triangular matrix. The decomposition is unique if $L$ has positive diagonal elements. As a result, the symmetric positive definite $\mathbf{M}$ can be fully and uniquely identified by $d(d+1)/2$ numbers, $d$ of which are constrained to be positive. We can then use a $d(d+1)/2$-dimensional normalizing flow to model probability densities over symmetric-positive matrices. We now illustrate how to transform the initial vector $\mathbf{z} \in \mathbb{R}^{d(d+1)/2}$ into a symmetric positive definite matrix $\mathbf{\Omega} \in \mathbb{R}^{d \times d}$. To the best of our knowledge, this is the first attempt to define a flow $\mathcal{T}(\mathbf{\Omega})$ on the space of symmetric positive definite matrices by construction.

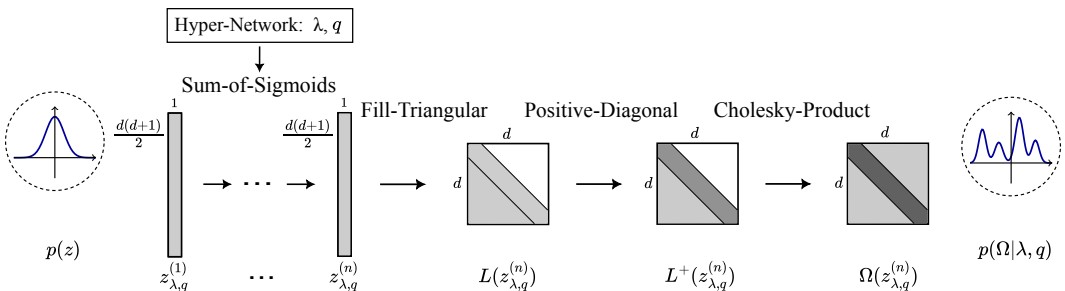

Figure 1: Architecture of the proposed Conditional Matrix Flow (CMF) model. Light grey is used to denote unconstrained values, dark grey for positive values, and white for zeros.

A high-level visualization of the proposed architecture is shown in Figure 1. The first $n$ layers of the network consist of arbitrary NF transformations and map the initial vector $\boldsymbol{z}$ to $\boldsymbol{z}_{\lambda,q}^{(n)}$. We propose to use a class of flexible transformations that we called Sum-of-Sigmoids, which worked particularly well for our applications. We provide further insight on the implementation in Appendix A.1. Furthermore, we condition the flow on $\lambda$ and $q$ via a hypernetwork. The hypernetwork takes $\lambda$ and $q$ as inputs and returns the parameters of the Matrix Flow. We transform $\boldsymbol{z}_{\lambda,q}^{(n)}$ into a symmetric positive definite matrix in three steps. First, the vector is raveled into a lower triangular matrix $\boldsymbol{L} \in \mathbb{R}^{d\times d}$, which we call *Fill-Triangular*. This transformation has a unit Jacobian determinant because it just reshapes the vector into a matrix. In a second step, we bijectively map the diagonal of the resulting lower triangular matrix to positive values with a softplus activation $\mathrm{Softplus}(\boldsymbol{x}) = \log(1 + \exp \boldsymbol{x})$. This transformation, which we call *Positive-Diagonal*, acts element-wise and hence admits a cheap Jacobian (log) determinant. Lastly, we compute the *Cholesky product* $\mathrm{Chol}(\boldsymbol{L}) : \boldsymbol{L} \mapsto \boldsymbol{L}\boldsymbol{L}^T$, which again has an inexpensive Jacobian (log) determinant [Gupta and Nagar, 1999]:

$$\det \mathcal{J}_{\mathrm{Chol}}(\boldsymbol{L}) = 2^d \prod_{i=1}^{d} (\boldsymbol{L}_{ii})^{d-i+1} \ , \tag{9}$$

which depends only on the $d$ diagonal elements of $\boldsymbol{L}$. Note that by enforcing the triangular matrix to be positive, we make sure that the Cholesky decomposition is unique, and hence a bijection.

We open-source the implementation of the conditional bijective layers.[1] We based our library, which was first used in Torres et al. [2023], on the high-level structure provided by the *nflows* library [Durkan et al., 2020].

### 3.4 Training through simulated annealing

With our CMF, we can perform posterior inference as a function of $\lambda$ and $q$ and to perform model selection. These are key advantages of the Bayesian perspective. In addition, we can also recover the frequentist solution path as a function of $\lambda$ without modifying the model. Specifically, we employ optimization through Simulated Annealing [Kirkpatrick et al., 1983] to approximately sample from the global maxima of the distribution, which for posteriors is the maximum a posteriori estimate (MAP). The idea comes from statistical mechanics, where slowly cooling processes are used to study the ground (optimal) state of the system. When applied to more general optimization tasks, simulated annealing consists in accepting iterative improvements if they lead to a lower cost function. Meanwhile, the temperature is slowly decreased from the initial value $T_0$ to $T_n \approx 0$, where the system is frozen and no further changes occur. In our setting, this corresponds to introducing an artificial temperature $T_i$ for the target posterior in Eq. (7):

$$p^{(i)}(\boldsymbol{\Omega}) = p(\boldsymbol{\Omega}|\boldsymbol{S}, \lambda, q)^{1/T_i} \ , \tag{10}$$

where $T_i$ is the temperature at the $i$-th iteration. If the initial temperature is high enough, $p^{(i)}(\boldsymbol{\Omega})$ will likely be very flat, allowing for a better exploration of the support of the distribution. As the temperature decreases, the distribution becomes more peaked. In the limit $T_n \to 0$ the distribution

---

[1]**FlowConductor**: (Conditional) Normalizing Flows and bijective Layers for Pytorch
`https://github.com/FabricioArendTorres/FlowConductor`

$p^{(i)}(\boldsymbol{\Omega})$ should concentrate on the global maximum, hence on the MAP solution. Geman and Geman [1984] formally showed that convergence to the set of global minima is guaranteed for logarithmic cooling schedules. Unfortunately, logarithmic cooling schedules are not viable in practice, so alternative schemes have been explored [Abramson et al., 1999]. In this paper, we use the popular geometric cooling schedule [Andrieu and Doucet, 2000, Yuan et al., 2004], which works well in practice: $T_i = T_0 a^{i/n}$ for some $a > 0$ and $i \in \{0, \ldots, n\}$. In practice, we train the objective in Eq. (8) while slowly cooling down the system, i.e. decreasing $T_i$ in Eq. (10), until we reach a sufficiently low temperature $T_n$. By saving the model at $T_i = 1$ and $T_n$ we have access to both the Bayesian and frequentist solutions, respectively.

### 3.5  Limitations

The proposed approach provides a general framework for posterior inference in GGMs and generalizes to a large family of likelihood and priors, which requires an efficient evaluation of their analytic expression. In some cases, e.g. for posterior predictive distributions, this would require an additional integration step, which could become computationally expensive. The proposed approach is also limited by the expressive power of the bijective layers of the flow. Even though state-of-the-art layers are extremely powerful in modeling high-dimensional distributions, each layer might still be limited in terms of the number of modes that can be modeled [Liao and He, 2021]. Lastly, the proposed model assumes that we can model a family of posterior distributions as a function of the conditioning parameters $q$ and $\lambda$, which ultimately depends on the flexibility of the hypernetwork used.

## 4  Experiments

In this section, we showcase the effectiveness of the proposed CMF first on artificial data and then on a real application. In particular, we study the evolution of the variational posterior as a function of $\lambda$ and $q$. We then perform model selection on $\lambda$ through marginal likelihood maximization. We further illustrate the effect of training through simulated annealing and show that we recover the frequentist solution path through the MAP. Lastly, we show that the proposed method can be readily applied to real data in higher-dimensional settings. Results show that sub-$l1$ pseudo-norms provide sparser solutions and contrast the well-known overshrinking effect of Lasso relaxation. We provide the code to reproduce the experiments at `https://github.com/marcello-negri/CMF/`.

### 4.1  Toy example: synthetic data

We illustrate how the proposed CMF works on artificially generated sparse precision matrices. The data generation process consists of sampling a sparse precision matrix [Pedregosa et al., 2011] with $d$ features and sparsity level $\alpha$, and then generating $n$ Gaussian samples accordingly. For illustrative purposes, we show results for one precision matrix with $\alpha = 90\%$ and $n = d = 15$ ($n = d$ ensures the invertibility of the empirical covariance). We trained our model for $10'000$ epochs through simulated annealing with an initial temperature of $T_0 = 5$ to $T_n = 0.01$ and performed 100 geometric cooling steps with $T_i = T_0 a^{i/n}$ for $a = T_n/T_0$.

Figure 2 shows the effect of simulated annealing on the posterior. At temperature $T = 1$ the conditional flow approximates the true posterior in Eq. (7), which for $q = 1$ coincides with the Bayesian Graphical Lasso (BGL) model. As we decrease the temperature, the (unnormalized) target posterior in Eq. (10) becomes more peaked and we observe shrinking credible intervals. At the final temperature $T_n$, the distribution converges to the MAP and we recover the frequentist path. As shown in Figure 2 (*right*), our model accurately reconstructs the solution path over $\lambda$ (MSE $= 0.052$). Furthermore, we perform model selection on $\lambda$ for the model in Eq. (7), which is obtained at temperature $T = 1$. In the Appendix A.4 in Figure 7, we show the (approximate) marginal log-likelihood as a function of $\lambda$ and the resulting optimal $\lambda^*_{\text{CMF}} = 3.52$ that maximizes it. We compare the result with the frequentist estimate obtained through MLE with 5-fold cross validation $\lambda^*_{\text{GLasso}} = 3.36$. Again, the model agrees with the frequentist solution.

One of the most significant drawbacks of relaxing the $l_0$-norm regularization with $l_1$ is that sparsity is achieved through shrinkage. This means that the selected relevant features are over-shrunk. With the proposed CMF we show that by exploring sub-$l_1$ pseudo-norms we can reduce the shrinkage effect and virtually overcome it in the $q \to 0$ limit. In Figure 3 (*left*) we show $95\%$ posterior

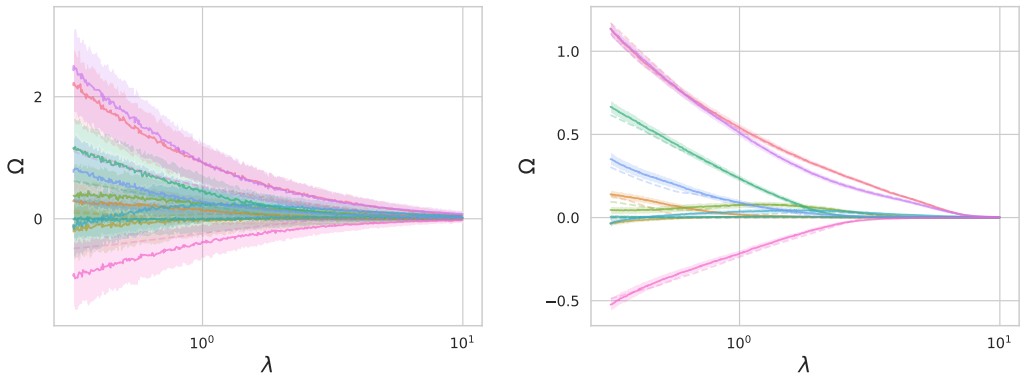

Figure 2: 75% posterior credible intervals as a function of $\lambda$ for $q = 1$. *Left:* at $T = 1.0$ the CMF reduces to the BGL. *Right:* at $T = 0.01$ the CMF reduces to the frequentist solution path (dashed).

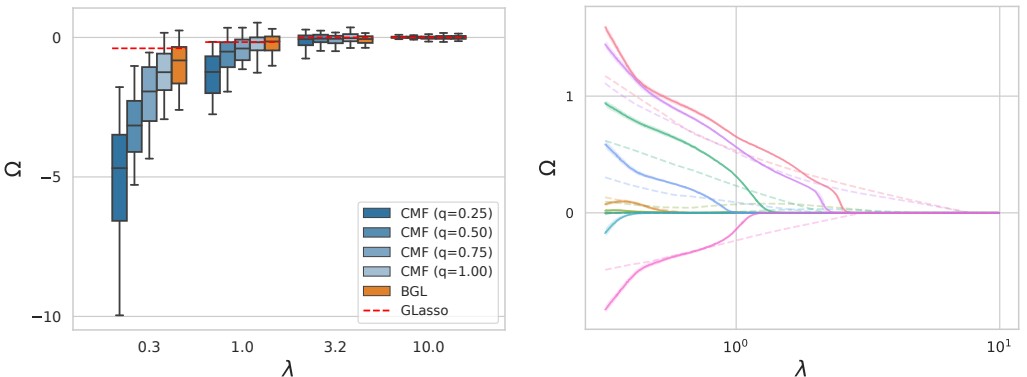

Figure 3: *Left:* 95% posterior credibility intervals of the proposed CMF for one entry of the precision matrix as a function of the pseudo-norm $q = \{1, 0.75, 0.5, 0.25\}$. Results are compared with the BGL and the frequentist solution (GLasso). *Right:* MAP estimate of the CMF posterior as a function of $\lambda$ for $q = 0.5$ ($T = 0.01$). The dashed line is the frequentist solution path for $q = 1$.

credibility intervals for one entry of the reconstructed precision matrix. We can observe that, as the $l_q$ pseudo-norm gets closer to $l_1$, the posterior median is progressively shrunk towards zero. With the proposed CMF we can prevent this shrinkage effect by exploring the posterior solution path for sub-$l_1$ pseudo-norms. In particular, note that for $\lambda = 0.3$ the posterior 95% credibility interval does not include the value 0 for $q = \{0.25, 0.5, 0.75\}$, as opposed to the BGL (or $q = 1$). In Figure 3 (*right*) we show the posterior solution path of the CMF for $q = 0.5$ in the MAP limit ($T = 0.01$). Compared to the $l_1$ solution path, sub-$l_1$ solution paths show less shrinkage effect (higher values for the selected features) and steepest decrease towards zero. This behaviour becomes more evident as $q$ decreases, as we show more extensively in the Appendix A.4 in Figure 6 for $q = \{1, 0.75, 0.5, 0.25\}$. These results support the interest in sub-$l_1$ pseudo-norms, and in $l_0$ norm in the limit.

## 4.2 Edge recovery with sub-$l_1$ pseudo-norms

We now study the behaviour of the proposed CMF in terms of F1 score for edge recovery as a function of the number of samples. We show that the proposed CMF outperforms competing methods, especially in the low sample regime. We compare the CMF with the BGL, which is the only alternative Bayesian model. Then, we compare the CMF in the $q \to 0$ limit against frequentist approaches that use approximate $l_0$ norm penalties: Atan ("atan") [Wang and Zhu, 2016], Seamless $l_0$ ("selo") [Dicker et al., 2013], log-penalty ("log") [Mazumder et al., 2011], SICA ("sica") [Lv and Fan, 2009].

We use 10 ground truth precision matrices of dimension $d = 30$ and generate $n$ Gaussian samples accordingly. We show results for the relevant $n < d$ regime and around $n = d$, namely for samples

$n = \{15, 25, 35, 45\}$. For Bayesian approaches (the proposed CMF and the BGL) we draw 1000 samples from the (approximate) posteriors and consider 90% credibility intervals. All results are averaged over the 10 precision matrices. The proposed CMF is trained for 5000 epochs to a final temperature $T = 1$, which corresponds to the Bayesian model in Eq. (7). For the BGL we run the Gibbs sampler for 4000 iterations with a burn-in of 1000 and keep every fourth sample. The frequentist approaches were run with the specific hyper-parameters suggested in their original papers.

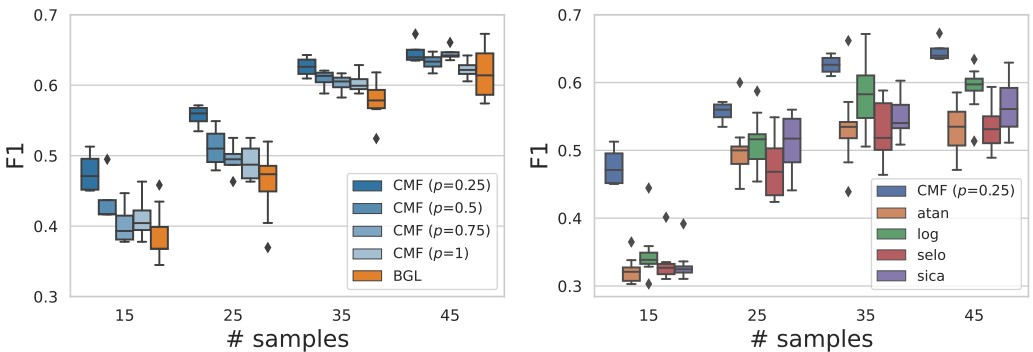

Figure 4: F1 score for edge recovery. *Left:* the proposed CMF for $q = \{0.25, 0.5, 0.75, 1\}$ against the Bayesian Graphical Lasso (BGL). *Right:* the proposed CMF for $q = 0.25$ against various frequentist approaches with penalties that approximate the $l_0$ norm. Results are averaged over 10 precision matrices with $d = 30$.

As the proposed method is inherently Bayesian we first compare it against the BGL. We show that for $q = 1$ we obtain results compatible with the BGL. Furthermore, with the proposed method we can additionally infer the posterior for $q < 1$. The results in Figure 4 (*left*) show that in the low sample regime ($n < d$) sub-$l_1$ pseudo-norms are beneficial and result in a higher F1 score. The effect is stronger as $q \to 0$, while in the $n > d$ regime sub-$l_1$ pseudo-norms do not provide a significant advantage. We also compare the proposed CMF with classical frequentist approaches with surrogate penalties that approximate the $l_0$ norm. The results in Figure 4 (*right*) show that the proposed CMF with $q = 0.25$ outperforms all competing methods across all sample regimes, but especially for very low sample sizes ($n = 15$). Note that in the $q < 1$ regime frequentist algorithms require an ad hoc initialization of the precision matrix, which we provided through the Ledoit-Wolf shrinkage estimator.

### 4.3 Real data application

We now consider a real-world application to showcase that the proposed CMF can be easily used in high-dimensional settings. We consider a setting in which we are interested in studying only a subset of query variables. In particular, we show that we can avoid inferring the posterior on the full precision matrix, which is expensive, by simply redefining the target posterior, i.e. the prior and likelihood. We consider a colorectal cancer dataset [Sheffer et al., 2009], which contains measurements of 7 clinical variables together with 312 gene measurements from biopsies for 260 cancer patients. As the dataset contains many missing values, we drop the *p53 mutation status* clinical variable and only consider $n = 190$ fully measured patients. We study the connections between the $s = 6$ clinical variables and the $t = 312$ gene expression measurements. Like Kaufmann et al. [2015], we consider the partition

$$\mathbf{\Omega} = \begin{pmatrix} \mathbf{\Omega}_{11} & \mathbf{\Omega}_{12} \\ \mathbf{\Omega}_{12}^T & \mathbf{\Omega}_{22} \end{pmatrix} \begin{matrix} s \\ t \end{matrix} \qquad \mathbf{S} = \begin{pmatrix} \mathbf{S}_{11} & \mathbf{S}_{12} \\ \mathbf{S}_{12}^T & \mathbf{S}_{22} \end{pmatrix} \begin{matrix} s \\ t \end{matrix} \qquad (11)$$

where $s \ll t$. Instead of inferring the expensive $(s + t) \times (s + t)$ precision matrix $\mathbf{\Omega}$, we study the $\mathbf{\Omega}_{11}$ and $\mathbf{\Omega}_{12}$ sub-matrices, which overall require only $s \times (s + t)$ dimensions. Following Kaufmann et al. [2015], we show in Appendix A.2 that we can infer the posterior $p(\mathbf{\Omega}_{11}, \mathbf{\Omega}_{12} | \mathbf{S}, \lambda)$ independently of the large $\mathbf{\Omega}_{22.1}$. But in contrast to Kaufmann et al. [2015], we (i) can enforce the correct double-exponential prior on $\mathbf{\Omega}_{11}$, we (ii) do not need to invert the $st \times st$ matrix, which takes $O(st^3)$ operations, and we (iii) do not require a Gibbs sampler, which can still suffer from poor mixing behaviour. In our framework, we only need to define a different posterior as the target

distribution. In practice, we define the flow jointly over the sub-matrices $\mathbf{\Omega}_{11}$, which must be positive definite, and $\mathbf{\Omega}_{12}$, which is unconstrained. It is then sufficient to define the bijective layers on the $s(s+1)/2$ plus $s \times t$ dimensions and to perform the Cholesky product only on the dimensions encoding $\mathbf{\Omega}_{11}$. Note that the determinant of the Jacobian of the full transformation is still the same as in Eq. (9) because the $s \times t$ dimensions encoding $\mathbf{\Omega}_{12}$ are just raveled into a matrix.

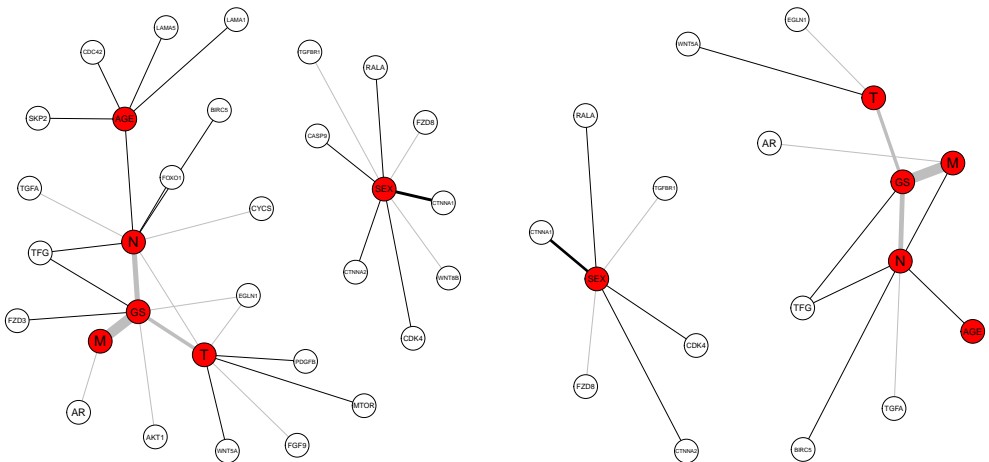

Figure 5: Inferred network structure with the proposed CMF for $q = 1.0$ (*left*) and $q = 0.6$ (*right*). Positive edges are shown in black and negative ones in grey. Clinical variables are highlighted in red.

In Figure 5 we show the inferred network structure with the proposed CMF for $q = 1.0$ (*left*) and for $q = 0.6$ (*right*). Significant edges are obtained by considering $80\%$ credible intervals on the posterior of $\mathbf{\Omega}_{11}$ and $\mathbf{\Omega}_{12}$. The clinical variables in the data are age, sex, cancer group stage (GS), and TNM classification for colorectal cancer, which measures its size (T), whether it spreads to lymph nodes (N) and if metastases develop (M). Note that, in contrast to [Kaufmann et al., 2015], we can infer the posterior on the $\mathbf{\Omega}_{11}$ block as well, which allows to study the conditional independence structure among clinical variables. For instance, results suggest strong connections between the cancer group stage (GS) and each variable of the TNM classification (T, N, M). More interestingly, with the proposed CMF we can infer the posterior for different sub-$l_1$ pseudo-norms. As expected, Figure 5 shows that with lower $q$ values we obtain sparser solutions. Additional quantitative comparisons for $q = \{1.0, 0.9, 0.8, 0.7, 0.6\}$ are included in the Appendix A.5 in Figure 8.

## 5    Conclusions

We propose a very general framework for variational inference in Gaussian Graphical Models through conditional normalizing flows. Our model unifies the benefits of Bayesian and frequentist approaches while avoiding most of their specific problems. Compared to existing approaches, the most important advantage of our method is that it can jointly train a continuum of sparse regression models for all regularization parameters and all $l_q$ (pseudo-) norms. All of these models can be analyzed both in a Bayesian fashion (at temperature $T = 1$) or, alternatively, in the frequentist limit (i.e. penalized likelihood) as $T$ approaches 0. To the best of our knowledge, this is the first Gaussian Graphical Model that can continuously explore all sparsity-inducing priors from the $l_q$-norm family. Moreover, thanks to our variational formalism, we can integrate out the model parameters and use the (approximate) marginal likelihood for model selection without any additional computational costs.

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

# A   Appendix

## A.1   Sum-of-Sigmoids layers

We implement the proposed model with a conditional flow architecture that relies on element-wise transformations through monotonic functions, which we term *Sum-of-Sigmoids*. We further combine these element-wise monotonic transformations in an autoregressive fashion using a MADE-like approach [Kingma et al., 2016, Huang et al., 2018]. The proposed layers are light and very flexible and work particularly well for our purposes already with only 4 layers. Note that, even though it is not needed in our framework, the inverse can be computed numerically, which is relatively cheap since the transformation is element-wise monotonic. Specifically, we implement flexible monotonic transformations by combining shifted and scaled sigmoid activations. Differently from Huang et al. [2018], we also add shifted (and flipped) softplus functions to the element-wise activations. This leads to a linear behaviour outside a specific range $[-s, s], s > 0$, alleviating the effect of inputs that lie outside the seen training data. The strictly monotonic element-wise transformation is given by

$$\phi_{sos}(z^{(i)}) = \underbrace{\left[ a \sum_{j=1}^{k} v_j \, \sigma(w_j z^{(i)} + b_j) \right]}_{\text{sum of monotonic functions}} + \underbrace{\left[ \ln \left( 1 + e^{(z^{(i)} - s)} \right) - \ln \left( 1 + e^{(-z^{(i)} - s)} \right) \right]}_{\text{approx. linear for } |z^{(i)}| \gg s, \text{ and zero for} |z^{(i)}| \ll s}, \quad (12)$$

where $\sigma$ is the sigmoid function and $\{v_j, w_j, b_j\}_{j=1}^{k}$ and $a$ are learnable parameters such that $w_k, v_j, a > 0$ and $\sum_{j=1}^{k} v_j = 1$. The advantage of using an element-wise transformation is that the associated Jacobian is diagonal and therefore its log-determinant is straightforward to compute. The proposed Sum-of-Sigmoids layers provide a very flexible transformation, but due to the element-wise nature, different dimensions are not mixed together. We overcome this by predicting the parameters $\{v_j, w_j, b_j\}_{j=1}^{k}$ and $a$ of the Sum-of-Sigmoids with masked autoregressive hypernetworks, similar to other masked autoregressive flows. When implemented through masking [Papamakarios et al., 2018], autoregressive flows allow for an elegant extension to conditional settings as well. Relevantly, this autoregressive structure admits a simple log-determinant Jacobian since the Jacobian is lower-triangular by construction.

## A.2   Joint posterior for $\mathbf{\Omega}_{11}$ and $\mathbf{\Omega}_{12}$

We show that given the partition in Eq. (11) and the prior in Eq. (7), the joint posterior factorizes as

$$p(\mathbf{\Omega}_{11}, \mathbf{\Omega}_{12}, \mathbf{\Omega}_{22.1} | \mathbf{S}, \lambda, q) = p(\mathbf{\Omega}_{11}, \mathbf{\Omega}_{12} | \mathbf{S}, \lambda, q) \, p(\mathbf{\Omega}_{22.1} | \mathbf{S}, \lambda, q) \,. \quad (13)$$

We further provide the analytic expression for $p(\mathbf{\Omega}_{11}, \mathbf{\Omega}_{12} | \mathbf{S}, \lambda, q)$. The latter is used as target distribution in the training loss in Eq. (8) for the real data application in Section 4. For this proof we followed the approach of Torres [2018].

Let $\mathbf{X} \in \mathbb{R}^{n \times (s+t)}$ be the design matrix containing independent observations. We are interested in estimating the connections between $s$ query variables with respect to the $t$ remaining ones, typically with $s \ll t$. Given the partition of $\mathbf{\Omega}$ and $\mathbf{S}$ in Eq. (11), we define the full posterior $p(\mathbf{\Omega}, \mathbf{S} | \lambda, q) = p(\mathbf{\Omega}_{11}, \mathbf{\Omega}_{12}, \mathbf{\Omega}_{22}, \mathbf{S} | \lambda, q)$ as the product of the Wishart likelihood $p(\mathbf{S} | \mathbf{\Omega})$ and a suitable prior $p(\mathbf{\Omega} | \lambda, q)$:

$$\begin{aligned} p(\mathbf{S} | \mathbf{\Omega}) &\propto \mathcal{W}_{s+t}(n, \mathbf{\Omega}^{-1}) = \det(\mathbf{\Omega})^{n/2} \, \exp\{\text{Tr}(-\tfrac{1}{2} \mathbf{\Omega} \mathbf{S})\} \\ p(\mathbf{\Omega} | \lambda, q) &\propto \mathcal{W}_{s+t}(s + t + 1, \lambda \mathbf{1}_{s+t}) \, p(\mathbf{\Omega}_{11} | \lambda, q) \, p(\mathbf{\Omega}_{12} | \lambda, q) \,. \end{aligned} \quad (14)$$

Specifically, we select a Wishart prior on $\mathbf{\Omega}$ to ensure positive definiteness on the full matrix. In order to encourage sparsity we choose a generalized Normal distribution prior over the off-diagonal elements of $\mathbf{\Omega}_{11}$ and over the full $\mathbf{\Omega}_{12}$:

$$p(\mathbf{\Omega}_{11} | \lambda, q) \propto \prod_{i<j} \exp \left( -\lambda |(\mathbf{\Omega}_{11})_{ij}|^q \right) \qquad p(\mathbf{\Omega}_{12} | \lambda, q) \propto \prod_{i,j} \exp \left( -\lambda |(\mathbf{\Omega}_{12})_{ij}|^q \right) \,. \quad (15)$$

As a first step we explicitly write down the expression for the likelihood

$$\begin{aligned} p(\mathbf{S} | \mathbf{\Omega}) &= p(\mathbf{S} | \mathbf{\Omega}_{11}, \mathbf{\Omega}_{12}, \mathbf{\Omega}_{22.1}) \\ &= \mathcal{W}_{s+t}(n, \mathbf{\Omega}^{-1}) \\ &\propto \det(\mathbf{\Omega})^{n/2} \det(\mathbf{S})^{(n-(s+t)-1)/2} \exp \left( -\tfrac{1}{2} \text{Tr} \left[ \mathbf{\Omega} \mathbf{S} \right] \right) \end{aligned} \quad (16)$$

and for the prior

$$
\begin{aligned}
p(\boldsymbol{\Omega}|\lambda, q) &= p(\boldsymbol{\Omega}_{11}, \boldsymbol{\Omega}_{12}, \boldsymbol{\Omega}_{22.1}|\lambda, q) \\
&\propto \mathcal{W}_{s+t}(s + t + 1, \lambda \mathbf{1}_{s+t})\, p(\boldsymbol{\Omega}_{11}|\lambda)\, p(\boldsymbol{\Omega}_{12}|\lambda) \\
&= \exp\left(-\tfrac{\lambda}{2}\mathrm{Tr}\big[\boldsymbol{\Omega}\big]\right) \prod_{i<j} \exp\left(-\lambda|(\boldsymbol{\Omega}_{11})_{ij}|^q\right) \prod_{i,j} \exp\left(-\lambda|(\boldsymbol{\Omega}_{12})_{ij}|^q\right).
\end{aligned}
\tag{17}
$$

Overall, the posterior reads as

$$
\begin{aligned}
p(\boldsymbol{\Omega}_{11}, \boldsymbol{\Omega}_{12}, \boldsymbol{\Omega}_{22.1}, \boldsymbol{S}|\lambda, q) &\propto \det(\boldsymbol{\Omega})^{n/2} \det(\boldsymbol{S})^{(n-(s+t)-1)/2} \\
&\quad \times \exp\left(-\frac{1}{2}\mathrm{Tr}\big[\boldsymbol{\Omega S} + \lambda \boldsymbol{\Omega}\big]\right) \\
&\quad \times \prod_{i<j} \exp\left(-\lambda|(\boldsymbol{\Omega}_{11})_{ij}|^q\right) \prod_{i,j} \exp\left(-\lambda|(\boldsymbol{\Omega}_{12})_{ij}|^q\right).
\end{aligned}
\tag{18}
$$

Following Kaufmann et al. [2015], we now re-write the posterior through the change of variables $(\boldsymbol{\Omega}_{11}, \boldsymbol{\Omega}_{12}, \boldsymbol{\Omega}_{22}) \to (\boldsymbol{\Omega}_{11}, \boldsymbol{\Omega}_{12}, \boldsymbol{\Omega}_{22.1})$ involving the Schur component $\boldsymbol{\Omega}_{22.1} = \boldsymbol{\Omega}_{22} - \boldsymbol{\Omega}_{12}^T \boldsymbol{\Omega}_{11}^{-1} \boldsymbol{\Omega}_{12}$. Note that the transformation has unit Jacobian: $J\big((\boldsymbol{\Omega}_{11}, \boldsymbol{\Omega}_{12}, \boldsymbol{\Omega}_{22}) \to (\boldsymbol{\Omega}_{11}, \boldsymbol{\Omega}_{12}, \boldsymbol{\Omega}_{22.1})\big) = \mathbf{1}$. We can now explicitly re-write the posterior as $p(\boldsymbol{\Omega}_{11}, \boldsymbol{\Omega}_{12}, \boldsymbol{\Omega}_{22.1}, \boldsymbol{S}|\lambda, q)$ by using the following substitutions, which are straightforward to show:

$$
\begin{aligned}
\det(\boldsymbol{\Omega}) &= \det(\boldsymbol{\Omega}_{11}) \det(\boldsymbol{\Omega}_{22} - \boldsymbol{\Omega}_{12}^T \boldsymbol{\Omega}_{11}^{-1} \boldsymbol{\Omega}_{12}) = \det(\boldsymbol{\Omega}_{11}) \det(\boldsymbol{\Omega}_{22.1}), \\
\mathrm{Tr}(\boldsymbol{\Omega S}) &= \mathrm{Tr}\big[\boldsymbol{\Omega}_{11}\boldsymbol{S}_{11} + \boldsymbol{\Omega}_{12}\boldsymbol{S}_{12}^T + \boldsymbol{\Omega}_{12}^T\boldsymbol{S}_{12} + \boldsymbol{\Omega}_{22.1}\boldsymbol{S}_{22} + \boldsymbol{\Omega}_{12}^T\boldsymbol{\Omega}_{11}^{-1}\boldsymbol{\Omega}_{12}\boldsymbol{S}_{22}\big], \\
\mathrm{Tr}(\lambda\boldsymbol{\Omega}) &= \mathrm{Tr}\big[\lambda\boldsymbol{\Omega}_{11} + \lambda\boldsymbol{\Omega}_{22.1} + \lambda\boldsymbol{\Omega}_{12}^T\boldsymbol{\Omega}_{11}^{-1}\boldsymbol{\Omega}_{12}\big].
\end{aligned}
\tag{19}
$$

By re-arranging the terms the posterior simplifies to:

$$
\begin{aligned}
p(\boldsymbol{\Omega}_{11}, \boldsymbol{\Omega}_{12}, \boldsymbol{\Omega}_{22.1}, \boldsymbol{S}|\lambda, q) &\propto \det(\boldsymbol{\Omega}_{11})^{n/2} \det(\boldsymbol{S})^{\frac{n-(s+t)-1}{2}} \\
&\quad \times \det(\boldsymbol{\Omega}_{22.1})^{n/2} \exp\left(-\frac{1}{2}\mathrm{Tr}\big[\boldsymbol{\Omega}_{22.1}(\boldsymbol{S}_{22} + \lambda\mathbf{1}_t)\big]\right) \\
&\quad \times \exp\left(-\frac{1}{2}\mathrm{Tr}\big[\boldsymbol{\Omega}_{11}(\boldsymbol{S}_{11} + \lambda\mathbf{1}_s) + 2(\boldsymbol{\Omega}_{12}^T\boldsymbol{S}_{12}) + \boldsymbol{\Omega}_{12}^T\boldsymbol{\Omega}_{11}^{-1}\boldsymbol{\Omega}_{12}(\boldsymbol{S}_{22} + \lambda\mathbf{1}_t)\big]\right) \\
&\quad \times \prod_{i<j} \exp\left(-\lambda|(\boldsymbol{\Omega}_{11})_{ij}|^q\right) \prod_{i,j} \exp\left(-\lambda|(\boldsymbol{\Omega}_{12})_{ij}|^q\right).
\end{aligned}
$$

If we now condition on $\boldsymbol{S}$, and specifically on $\boldsymbol{S}_{22}$, we can clearly see that the posterior factorizes as $p(\boldsymbol{\Omega}_{11}, \boldsymbol{\Omega}_{12}, \boldsymbol{\Omega}_{22.1}|\boldsymbol{S}, \lambda, q) \propto p(\boldsymbol{\Omega}_{11}, \boldsymbol{\Omega}_{12}|\boldsymbol{S}, \lambda, q)\, p(\boldsymbol{\Omega}_{22.1}|\boldsymbol{S}, \lambda, q)$. In particular, we are interested in estimating the joint posterior $p(\boldsymbol{\Omega}_{11}, \boldsymbol{\Omega}_{12}|\boldsymbol{S}, \lambda, q)$, which reads as

$$
\begin{aligned}
p(\boldsymbol{\Omega}_{11}, \boldsymbol{\Omega}_{12}|\boldsymbol{S}, \lambda, q) &\propto \det(\boldsymbol{\Omega}_{11})^{n/2} \\
&\quad \times \exp\left(-\frac{1}{2}\mathrm{Tr}\big[\boldsymbol{\Omega}_{11}(\boldsymbol{S}_{11} + \lambda\mathbf{1}_s) + 2(\boldsymbol{\Omega}_{12}^T\boldsymbol{S}_{12}) + \boldsymbol{\Omega}_{12}^T\boldsymbol{\Omega}_{11}^{-1}\boldsymbol{\Omega}_{12}(\boldsymbol{S}_{22} + \lambda\mathbf{1}_t)\big]\right) \\
&\quad \times \prod_{i<j} \exp\left(-\lambda|(\boldsymbol{\Omega}_{11})_{ij}|^q\right) \prod_{i,j} \exp\left(-\lambda|(\boldsymbol{\Omega}_{12})_{ij}|^q\right)
\end{aligned}
\tag{20}
$$

### A.3   Run-time comparison with Gibbs sampler

The proposed CMF provides a significant speed-up in terms of sampling with respect to standard Gibbs sampling algorithms for posterior inference in Gaussian Graphical Models. For a realistic and fair comparison, we measure run-time on our real data experiment. For this purpose, we trained the Conditional Matrix Flow for $q = 1$ and compare its sampling speed against the Gibbs sampler introduced in Kaufmann et al. [2015]. The setting presented here is the same one used to obtain the results shown in the Experiment section. The Gibbs sampler takes 89 seconds to generate 500 samples, which translates to about 5.6 samples per second. This result was obtained on a Intel(R) Xeon(R) CPU E5-1660 v3 @ 3.00GHz. On the ohter hand, on the consumer-grade GPU NVIDIA TITAN X (12GB VRAM) sampling is extremely efficient: each second we can generate 2000 independent samples from the approximate posterior. Relevantly, in order to retrieve the full posterior path as a function of $\lambda$, we would need independent Markov Chains for each $\lambda$ value, rendering the approach infeasible. In contrast, with the proposed framework we can draw independent samples for different $\lambda$ values at the same computational cost.

## A.4 Artificial data

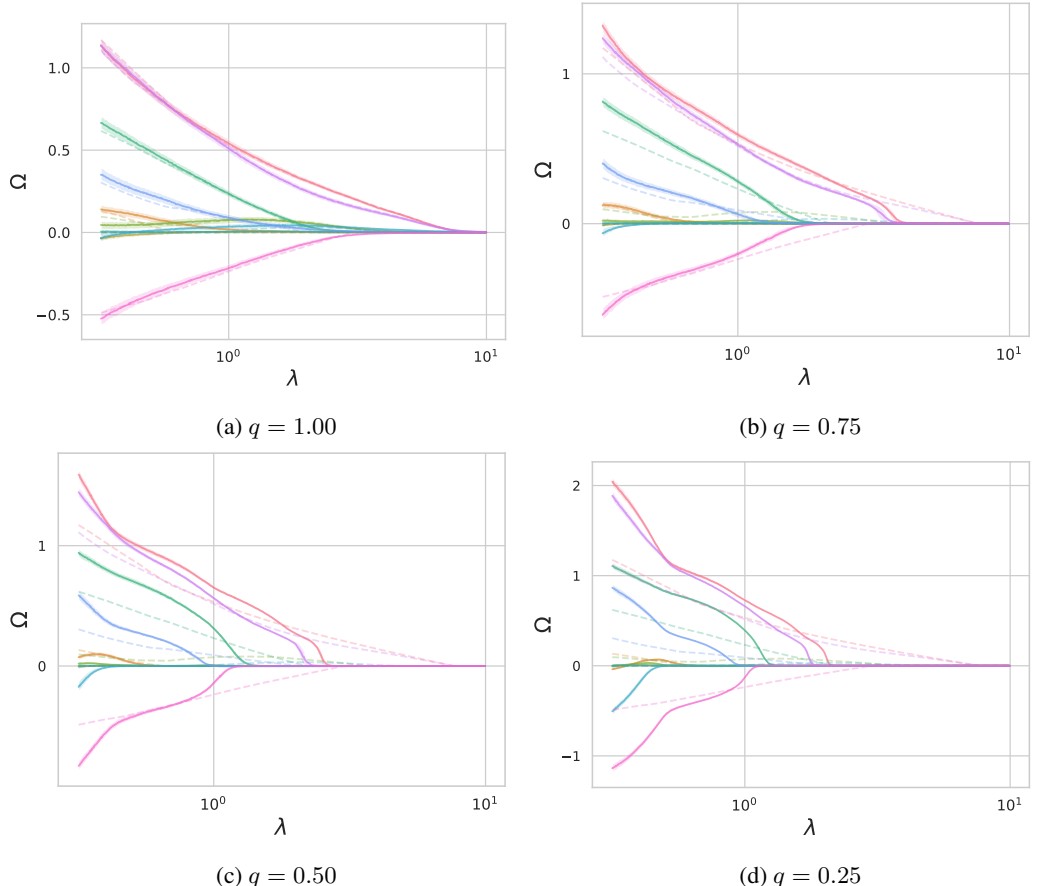

(a) $q = 1.00$

(b) $q = 0.75$

(c) $q = 0.50$

(d) $q = 0.25$

Figure 6: MAP estimate as a function of $\lambda$ for different sub-$l_1$ pseudo-norms. The dashed line is the frequentist solution path for $q = 1$.

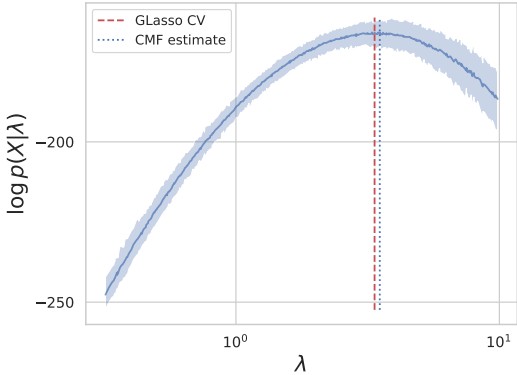

Figure 7: Evolution of the (approximate) marginal log-likelihood of the CMF as a function of $\lambda$ for $T = 1.00$ and $q = 1$. We select the optimal $\lambda$ as its maximum ($\lambda_{\mathrm{CMF}} = 3.52$, in blue) and compare with the frequentist estimate obtained through cross validation ($\lambda_{\mathrm{GLasso}} = 3.36$, in red).

## A.5 Real data

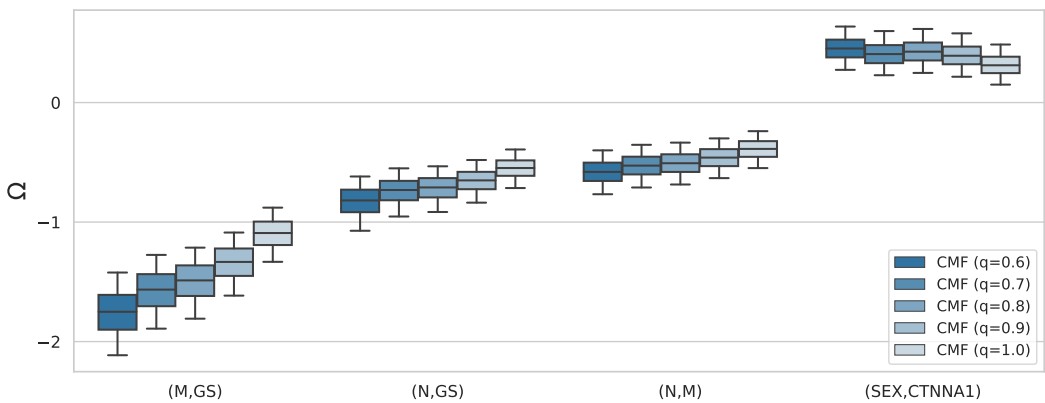

Figure 8: 95% posterior credibility intervals of the proposed CMF for the 4 entries with highest absolute median posterior as a function of the pseudo-norm $q = \{1, 0.9, 0.8, 0.7, 0.6\}$.

