# OpenReview forum: "Conditional Matrix Flows for Gaussian Graphical Models"
_NeurIPS.cc/2023/Conference — NeurIPS 2023 poster_

### Official Review · Reviewer_vqBu · 2023-06-27

**Soundness:** 3 good
**Presentation:** 3 good
**Contribution:** 3 good
**Rating:** 6
**Confidence:** 3

**Summary:**

This work proposed a framework for performing inference on Gaussian Graphical Models by approximating the posterior with a normalizing flow over PSD matrices. In this way, the authors can investigate $l_p$-norm regularized GGMs for any value of $p$ in an efficient way.

**Strengths:**

The idea of using normalizing flows for GGM inference definitely brings in advantages of both Bayesian and frequentist worlds; to me, that's an innovative idea.

**Weaknesses:**

The main weakness that I identified is the lack of comparison between the proposed framework and the well-studied graphical lasso with concave approximations of the $l_0$-norm. More precisely, the authors show that their framework obtains frequentist solution paths through simulated annealing, therefore, it'd be of great interest to see a comparison between these solution paths and those obtained by iterative algorithms such as iterative reweighted l1-norm for graphical lasso.

**Questions:**

- in the frequentist case, how does the proposed framework compares against more classical techniques to obtain the solution paths, e.g., iterative reweighted l1-norm?

**Limitations:**

The authors adequately addressed the limitations of their proposed framework.

---

> ### Author Rebuttal · Authors · 2023-08-09
>
> We thank the reviewer for acknowledging the novelty of the proposed framework and for suggesting relevant experiments, which we have included in the global response. We hope to provide satisfying answers to the concerns raised.
>
> - "The main weakness that I identified is the lack of comparison between the proposed framework and the well-studied graphical lasso with concave approximations of the $l_0$ norm."
>
> In the global response we provide further experiments to compare the proposed method against several frequentist approaches with penalties that approximate the $l_0$ norm. We include Atan by Wang and Zhu (2016), Seamless L0 by Dicker et al. (2013), Log by Mazumder et al. (2011) and SICA by Lv and Fan (2009). Overall, results show that the proposed method outperforms all the considered frequentist approaches in terms of F1 score for structural recovery.
>
> - "More precisely, the authors show that their framework obtains frequentist solution paths through simulated annealing, therefore, it'd be of great interest to see a comparison between these solution paths and those obtained by iterative algorithms such as iterative reweighted l1 norm for graphical lasso. Questions: in the frequentist case, how does the proposed framework compares against more classical techniques to obtain the solution paths, e.g., iterative reweighted l1-norm?"
>
> In the global answer we provide an experimental comparison specifically for an iterative re-weighted lasso approach, namely the Adaptive Lasso by Zou (2006). Results show that the proposed method consistently outperforms Adaptive lasso over a range of $l_p$ norms and across all number of samples. Furthermore, we compare our model against two other frequentist approaches with penalties that allow to interpolate between $l_1$ and $l_0$ norms: SCAD by Fan and Li (2001) and MC+ by Zhang (2010). Also in this case we outperform competing methods. For lack of space we have not included these results in the 1-page PDF. However, they can be reproduced by running the code linked in the rebuttal.

---

> > ### Comment · Reviewer_vqBu · 2023-08-17
> > **ack**
> >
> > Thank you for your efforts putting forth this rebuttal. I’ll keep my score as is.

---

### Official Review · Reviewer_3sWQ · 2023-07-03

**Soundness:** 2 fair
**Presentation:** 3 good
**Contribution:** 2 fair
**Rating:** 5
**Confidence:** 3

**Summary:**

This paper proposes a method that can be used to infer conditional independencies in a Gaussian model. These conditional independencies are related to zeros in the precision matrix. Typically, sparse enforcing norms are used to estimate the precision matrix while enforcing zeros in the elements outside of the diagonal. In this paper a Bayesian approach is considered. For this a pseudo-distribution for the data is considered by taking the exponential to the p-norm. The method is trained via variational inference combined with normalizing flows to increase the accuracy of the posterior approximation. The variational distribution is tuned via simulated annealing and a temperature parameter allows to interpolate between the Bayesian and the Map solution.

**Strengths:**

- Well written paper.

        - Illustrative toy experiments.

**Weaknesses:**

- The proposed method is a combination of already known techniques.

        - The experimental section is weak as only a single real problem is considered.

        - Although the proposed method is a generalization of several known techniques, I have found in the experimental section a lack of comparisons with other related methods.

        My main point of criticism is the weak experimental section which only considers a single real problem and no comparisons with other related methods are carried out in real problems.

        Another point of criticism is that, for some particular values of the p parameter one does not actually observe sparsity in the Bayesian solution. For example, when sampling from the Laplace distribution one never observes zeros in practice. Spike and slab priors (a mix between a Gaussian and a point of mass center at zero) are the ones that actually lead to zeros.

**Questions:**

None

**Limitations:**

The authors have not commented on the limitations of their approach.

---

> ### Author Rebuttal · Authors · 2023-08-09
>
> We thank the reviewer for the useful feedback on the experiment section and for acknowledging the clarity in the presentation of the proposed approach. We hope to provide satisfying answers to the concerns raised.
>
> - "The proposed method is a combination of already known techniques."
>
> We believe that one of the benefits of the proposed approach is indeed to provide a general framework that unifies the two known approaches, namely the Bayesian and frequentist ones. We do so by combining their advantages: we have access to the marginal likelihood for model selection, we can compute credible intervals and we can recover frequentist solution paths. However, we also improve upon existing methods. As a key advantage to current approaches, we (i) are able to model the posterior in the sub-$l_1$ norm regime and we (ii) have access to a continuum of Bayesian models as a function of both $\lambda$ and $p$. In the global response we (iii) show quantitatively that the proposed approach outperforms both the Bayesian Graphical Lasso and several frequentist approaches in terms of graphical structure recovery.
>
> - "The experimental section is weak as only a single real problem is considered"
>
> In the paper we considered one real problem to showcase that the proposed method can be readily applied to high dimensional real world settings. As part of the rebuttal, we decided to focus on further synthetic experiments in order to have access to the ground truth for a quantitative comparison with competing methods.
>
> - "Although the proposed method is a generalization of several known techniques, I have found in the experimental section a lack of comparisons with other related methods"
>
> For a more detailed comparison with Bayesian and frequentist approaches, we provide further experiments in the global response.
>
> - "for some particular values of the p parameter one does not actually observe sparsity in the Bayesian solution. For example, when sampling from the Laplace distribution one never observes zeros in practice. Spike and slab priors (a mix between a Gaussian and a point of mass center at zero) are the ones that actually lead to zeros"
>
> Contrary to frequentist approaches, Bayesian approaches provide credible intervals on the solution. As correctly pointed out, this implies that individual samples may not contain exact zeros. Indeed, this is precisely what motivates our interest in studying solutions in the sub-$l_1$ pseudo-norms, which lead to exact zeros in the $p\rightarrow 0$ limit.
>
> - "The authors have not commented on the limitations of their approach"
>
> For a discussion on the limitations of the proposed model we refer to the global response.

---

### Official Review · Reviewer_pTVu · 2023-07-07

**Soundness:** 3 good
**Presentation:** 4 excellent
**Contribution:** 2 fair
**Rating:** 6
**Confidence:** 3

**Summary:**

This paper targets the structure learning problem in Gaussian Graphical Models via (Normalising) Flow-based Variational approximation of the elements of weight metrics that correspond to the Gaussian Bayesian network.
They use sub-l1 pseudo norms to penalize dense precision metrics (which correspond to graphs with numerous links) without imposing an extra high penalty for large non-zero values (which typically occurs if $l_{\geq1}$ is used).

**Strengths:**

1. Up to my knowledge, this is the first time flows are applied to the space of positive definite matrices.
2. The proposed approach is flexible meaning the class of applicable prior and likelihood functions is quite large.
3. Using sub-l1 norm is suitable for structure learning.
4. The proposed algorithm is mathematically sound (as far as I can follow) and is quite interesting.
5. The paper is well-written, and the relevant work is sufficiently discussed.
6. Due to its flexibility, the proposed method has the potential of having a large impact.

**Weaknesses:**

Due to the factors mentioned in the previous section, I find this work impressive and beautiful. However, unfortunately, the carried out experiments are minimal. Most notably, the algorithm is compared to no alternative work (neither in the main paper nor in the supplementary material). With no quantitative comparisons, it is impossible to evaluate the performance of the proposed algorithm compared to the existing methods.

NOTE: In the Rebuttal, some experiments are carried out (though the code is still not accessible).

Minor suggestion:
1. Though it is clear in the context, I suggest that the authors do not use the same letter "p" (with the same font) for both probability density and norm parameter.
2. Fix minor typos e.g. the end sentence period in line 214.

**Questions:**

* In line 141, what do you mean by "contradiction"?

**Limitations:**

The authors should compare their method with the relevant structure learning lierature and reveal its points of strength as well as its limitations.

This work is theoretical/methodological and does not have any positive or negative social/ethical impact on its own.

---

> ### Author Rebuttal · Authors · 2023-08-09
>
> We thank the reviewer for the very encouraging comments and for acknowledging the potentially large impact of the proposed model. We hope to provide satisfying answers to the concerns raised.
>
> - "Most notably, the algorithm is compared to no alternative work (neither in the main paper nor in the supplementary material). With no quantitative comparisons, it is impossible to evaluate the performance of the proposed algorithm compared to the existing methods. Due to weak evaluation, I do not think this work is ready for publication except if the authors can provide sufficient quantitative experimental results (along with code) in the rebuttal."
>
> For a more detailed comparison with both frequentist and Bayesian approaches, we provide further experiments in the global response. We provide quantitative comparison in terms of accuracy in graphical structure learning and we compare the presented approach with the Bayesian Graphical Lasso and several frequentist method. In the Bayesian setting we recover BGL results and obtain higher F1 scores in the sub $l_1$ regime ($p<1$). The presented approach outperforms also classical frequentist algorithms across the sub-$l_1$ pseudo-norm and in the $l_0$ limit as well. We provide the code to reproduce our results as a comment to the AC.
>
> - "Though it is clear in the context, I suggest that the authors do not use the same letter 'p' (with the same font) for both probability density and norm parameter."
>
> Our notation was indeed confusing. We now changed it to $l_q$ norm and kept $p$ for probability density.
>
> - "In line 141, what do you mean by 'contradiction'?"
>
> Kaufmann et al. (2015) showed how to efficiently infer the posterior on the blocks $\Omega_{11}$ and $\Omega_{22}$ independently of the large block $\Omega_{22}$. However, in order to derive the algorithm for the Gibbs sampler, they drop the sparsity prior on the $\Omega_{11}$ block. In our opinion, this creates a contradiction in the model because the resulting model fails to enforce sparsity on the $\Omega_{11}$ block.
>
> - "The authors should compare their method with the relevant structure learning literature and reveal its points of strength as well as its limitations."
>
> In the global response, we provided further experiments specifically to evaluate structure learning and compare the proposed approach with both the Bayesian Graphical Lasso and several frequentist approaches. For a discussion on the limitations of the proposed model we refer to the global response.

---

### Official Review · Reviewer_CnQu · 2023-07-07

**Soundness:** 3 good
**Presentation:** 3 good
**Contribution:** 3 good
**Rating:** 6
**Confidence:** 4

**Summary:**

This paper concerns the estimation of precision matrix under $l_p$ norm sparcity penal. The solution is a variational inference through normalizing flow, which is a function of shrinkage parameter $\lambda$ and non-negative norm parameter $p$. It allows for straightforward computation of solution paths for the intervals of $\lambda$ and $p$, and was empirically evaluated on two relatively small data sets.

**Strengths:**

Framework for GGM estimation based on conditional normalizing flows, indeed appears novel. Supporting math seems solid.

Using simulated annealing algorithm to recover a path of solutions for varying $\lambda$ and $p$ is useful, in particular for the case of $p$, as in case of $\lambda$ it was fairly straightforward to perform it with other methods too. I am just wondering how costly and scalable it is under the new framework, an empirical/theoretical analysis would be appreciated.

**Weaknesses:**

Empirical evaluation appears limited. It does not contain comparison with other (e.g. frequentist) approaches to derive the solution paths. Both in terms of estimation accuracy and in terms of computational cost.

**Questions:**

In synthetic data example, why did you choose to have more samples than dimensions ( $n>d$ )? Since in that case GGM can be obtained with matrix inverse, and no need for penalized objective.

**Limitations:**

Limitations were not discussed.

---

> ### Author Rebuttal · Authors · 2023-08-09
>
> We thank the reviewer for acknowledging the novelty and soundness of our work. We hope to provide satisfying answers to the concerns raised.
>
> - "Using simulated annealing algorithm to recover a path of solutions for varying $\lambda$ and $p$ is useful, in particular for the case of $p$, as in case of $\lambda$ it was fairly straightforward to perform it with other methods too. I am just wondering how costly and scalable it is under the new framework, an empirical/theoretical analysis would be appreciated."
>
> The proposed approach is particularly suitable to study the evolution of the posterior both as a function of $\lambda$ and $p$ (and of the two together). Bayesian Graphical Lasso allows to infer the posterior only for $p=1$ and requires to re-start the Markov Chain for every different $\lambda$ value. Clearly, this becomes unfeasible very quickly. In contrast, posterior inference with the proposed CMF can be performed in few seconds on consumer-grade GPUs. The only computational cost is thus training, which however can be performed in around 10 minutes. Frequentist approaches are indeed faster and can compute solution paths in less than a minute. However, this is not surprising since they only recover a single point-wise path, while the presented CMF models a (infinite) family of posterior distributions, conditioned on both $\lambda$ and $p$. It is however relevant that as a particular instance of our model we can recover the frequentist solution paths.
>
> - "Empirical evaluation appears limited. It does not contain comparison with other (e.g. frequentist) approaches to derive the solution paths. Both in terms of estimation accuracy and in terms of computational cost"
>
> For a more detailed comparison with frequentist (and Bayesian) approaches in terms of estimation accuracy, we provide further experiments in the global response. In terms of computational cost we refer to the previous answer.
>
> - "In synthetic data example, why did you choose to have more samples than dimensions ($n>d$)? Since in that case GGM can be obtained with matrix inverse, and no need for penalized objective"
>
> We illustrated the presented model for $n>d$ just as a toy experiment. This is similar to the experiments in the original BGL paper by Wang et al (2012), where they consider $(d=30, n=50)$ and $(d=100, n=200)$. However, we agree that the relevant range is indeed $n<d$. In the global answer we included further experiments to specifically address this regime.
>
> - "Limitations were not discussed"
>
> For a discussion on the limitations of the proposed model we refer to the global response.

---

### Author Rebuttal · Authors · 2023-08-09

## Global response
We would like to thank all reviewers for the very encouraging comments and for acknowledging the novelty and the potentially large impact of the proposed approach. As all reviewers expressed one main concern, we try to address it in the global response. The main point of criticism concerns the experimental section and in particular the lack of comparisons with classical frequentist approaches for Gaussian Graphical Models (GGMs). As pointed out by reviewers vqBu and 3sWQ, the relevance and novelty of the proposed method is to unify the benefits of the Bayesian and frequentist approaches. However, as a key advantage on current Bayesian methods, the proposed method enables us to additionally explore all sparsity-inducing priors from the sub $l_1$ norm family. Even though the proposed method is inherently Bayesian, we agree that it is indeed interesting to compare its behaviour with the frequentist approaches. As a sanity check, in the paper we showcased that the proposed model recovers the frequentist solution path in the annealed limit. In the hope of resolving the concerns raised, we provide three additional experiments to compare the proposed approach with Bayesian and frequentist methods. As suggested, we study the performance in graphical structure learning (reviewer pTVu) in the $n<d$ regime (reviewer CnQu) and we focus on penalties that approximate the $l_0$ norm and sub-$l_1$ pseudo-norms (reviewer vqBu).

### Further experiments
We show that in the in the sub-$l_1$ regime the proposed Conditional Matrix Flow approach (CMF) outperforms both Bayesian Graphical Lasso (BGL) and the classical frequentist approaches.  In particular, we measure the F1 score for edge recovery as a function of the number of samples.  We first compare the CMF with the BGL, which is the only alternative Bayesian model. Then, we compare the CMF in the $p\rightarrow0$ limit against the frequentist approaches with approximate $l_0$ norm penalties: Atan ("atan") by Wang and Zhu (2016), Seamless L0 ("selo") by Dicker et al. (2013), Log ("log") by Mazumder et al. (2011), SICA ("sica") by Lv and Fan (2009).  Finally, we compare the CMF solution as a function of $p$ against frequentist approaches with penalties that allow to interpolate between $l_1$ norm and $l_0$ norm: SCAD ("scad") by Fan and Li (2001), MC+ ("mcp") by Zhang (2010) and Adaptive lasso ("adapt") by Zou (2006).

We use 10 ground truth precision matrices of dimension $d=30$ and generate $n$ Gaussian samples accordingly. For illustrative purposes we report results for the relevant regime $n<d$ and around $n=d$, namely for samples $n=\{15, 25, 35, 45\}$. More exhaustive results can be obtained by running the provided code. For Bayesian approaches (the proposed CMF and BGL) we draw 1000 samples from the (approximate) posteriors and consider $90\%$ credibility intervals. All results are averaged over the 10 precision matrices. The proposed CMF is trained for 5000 epochs to a final temperature $T=1$, which corresponds to the Bayesian model in Eq. (9). For the BGL we run the Gibbs sampler for 4000 iterations with a burn-in of 1000. We thin the samples and keep every fourth. The frequentist approaches were run each with the specific hyper-parameters suggested in their original papers.

1.  As the proposed method is inherently Bayesian we first compare it against the BGL. We show that for $p=1$ we recover the BGL results. Furthermore, with the proposed method we can additionally infer the posterior for $p<1$. The results in Figure 1 show that in the low sample regime ($n<d$) sub $l_1$-pseudo norms are beneficial and result in higher F1 score. The effect is stronger as $p \rightarrow 0$, while in the $n>d$ regime sub-$l_1$ pseudo-norms do not provide a significant advantage.
2. We compare the proposed CMF also with classical frequentist approaches with surrogate penalties that approximate the $l_0$ norm. Results in Figure 2 show that the proposed CMF with $p=0.25$ outperforms all competing methods across all number of samples, especially for very low sample sizes ($n=15$). Note that in the $n<d$ regime frequentist algorithms require an ad-hoc initialization of the precision matrix, which we provided through the Ledoit-Wolf shrinkage estimator.
3. Lastly, we compare the CMF against frequentist approaches with penalties that allow to smoothly interpolate between the $l_1$ and $l_0$ norms as a function of the parameter $\gamma$. We evaluate the proposed CMF for $p=\{0.25,0.5,0.75,1\}$. Results show that we outperform all baseline methods. In Figure 3 we illustrate the results for the best baseline approach, namely Adaptive Lasso. Both the CMF and the Adaptive Lasso show increased performance with more regularization (for $p \rightarrow 0$ and $\gamma \rightarrow 0$) but the CMF still provides a higher F1 score across all number of samples. Results for SCAD and MC+ can be obtained by running the provided code.

The link to the code for the experiments can be found in a separate comment to the AC.

### Limitations
The proposed approach provides a general framework for posterior inference in GGMs and generalizes to a large family of likelihood and priors, which requires to efficiently evaluate their analytic expression. In some cases, e.g. for posterior predictive distributions, this would require an additional integration step, which could become computationally expensive. The proposed approach is also limited by the expressive power of the bijective layers of the flow. Even though state-of-the-art layers are extremely powerful in modelling high-dimensional distributions, each layer might still be limited in terms of the number of modes that can be modeled (see Liao et al (2021)). Lastly, the proposed model assumes that we can model a family of posterior distributions as a function of the conditioning parameters $\lambda$ and $p$, which ultimately depends on the flexibility of the hyper-network used.

---

### Decision · Program_Chairs · 2023-09-21

**Decision:**

Accept (poster)

**Comment:**

This paper introduces a novel approach for inferring conditional independencies within a Gaussian model. These independencies correspond to the presence of zeros in the precision matrix. While conventional techniques involve employing norms that encourage sparsity to estimate the precision matrix while maintaining zeros off the diagonal, this study adopts a Bayesian perspective.

In this context, a pseudo-distribution is constructed for the data by exponentiating the p-norm. The method leverages variational inference along with normalizing flows to enhance the accuracy of the posterior approximation. The variational distribution is fine-tuned using simulated annealing, and a temperature parameter facilitates interpolation between the Bayesian and Maximum A Posteriori (MAP) solutions.

A particularly innovative aspect is the framework for estimating Gaussian Graphical Model (GGM) using conditional normalizing flows. The utilization of a simulated annealing algorithm allows for the exploration of a solution path distribution, contrasting the single path typical of frequentist approaches. Although an initial concern about limited experimental validation was raised, the authors addressed this issue during the rebuttal phase.

Given these insights, I recommend the acceptance of this paper.